# Tuberculosis Genetic Epidemiology: A Latin American Perspective

**DOI:** 10.3390/genes10010053

**Published:** 2019-01-16

**Authors:** Marc Woodman, Ilsa L. Haeusler, Louis Grandjean

**Affiliations:** 1Institute of Child Health, University College London, London WC1N 3JH, UK; marcjwoodman@gmail.com (M.W.); i.haeusler@ucl.ac.uk (I.L.H.); 2Department of Medicine, Imperial College London, London W2 1NY, UK; 3Great Ormond Street Hospital, Institute of Child Health, University College London, London WC1N 3JH, UK; 4Laboratorio de Investigacion y Desarollo, Universidad Peruana Cayetano Heredia, Av. Honorio Delgado 430, San Martin de Porres 15102, Lima, Peru

**Keywords:** *Mycobacterium tuberculosis*, tuberculosis, multidrug-resistant tuberculosis, *Mycobacterium tuberculosis* complex, genetics, whole-genome sequencing, South America, Latin America, pathology, transmission

## Abstract

There are an estimated 10 million new cases of tuberculosis worldwide annually, with 282,000 new or relapsed cases each year reported from the Americas. With improvements in genome sequencing technology, it is now possible to study the genetic diversity of tuberculosis with much greater resolution. Although tuberculosis bacteria do not engage in horizontal gene transfer, the genome is far more variable than previously thought. The study of genome-wide variation in tuberculosis has improved our understanding of the evolutionary origins of tuberculosis, the arrival of tuberculosis in Latin America, the genetic determinants of drug resistance, and lineage-specific associations with important clinical phenotypes. This article reviews what is known about the arrival of tuberculosis in Latin America, the genetic diversity of tuberculosis in Latin America, and the genotypic determinants of clinical phenotypes.

## 1. Introduction

*Mycobacterium tuberculosis* (MTB), the bacterial pathogen which causes tuberculosis disease (TB), is the leading cause of death from a single infectious agent worldwide, as well as being one of the overall top ten causes of death [1]. In 2017, it was estimated that 10 million people developed TB, leading to 1.3 million deaths [1]. Worldwide, the incidence rate of tuberculosis fell by 2% annually between 2000 and 2017; however, an estimated reduction of 4–5% per year is necessary to reach the first milestone of the END TB strategy by 2020 [1]. It is estimated that 1.7 billion people, about 23% of the world’s population, have latent TB infection and are therefore at risk of developing active TB disease throughout their lifetime [1]. Furthermore, the crisis of drug-resistant tuberculosis poses a growing threat to global TB control, with over half a million new drug-resistant cases reported in 2017.

Novel treatments and control strategies are therefore urgently required in order to make greater and more rapid reductions in tuberculosis incidence. New treatments such as bedaquiline [2], delamanid [3], and pretomanid [4] offer therapeutic potential in multidrug-resistant tuberculosis (MDR-TB). Injectable-free regimes for MDR-TB will hopefully improve tolerance, compliance, and cure rate. Improvements in whole-genome sequencing technology have allowed human and pathogen genomes to be studied with unprecedented definition at ever-increasing speed and lower cost. Sequencing studies in tuberculosis have already elucidated evolutionary pathways [5], improved diagnosis [6], and tuberculosis transmission epidemiology [7].

This article will review trends in Latin American tuberculosis and will focus on how the study of MTB genetic epidemiology has improved the understanding of the evolution, epidemiology, and pathogenesis of tuberculosis disease in the region.

## 2. Discussion

### 2.1. The Epidemiology of Tuberculosis in Latin America

Overall, 3% of global tuberculosis cases are reported from the Americas with an associated mortality rate of 7.3% [8]. In 2017, the incidence rate of tuberculosis in South America was 46.2 per 100,000 people, with corresponding values from the Caribbean and Central America (including Mexico) of 61.2 and 25.9 per 100,000 respectively. The epidemiology of tuberculosis varies considerably between countries within Latin America. Three countries—Brazil, Peru, and Mexico—account for slightly more than half of all cases in the Americas [8]. Peru has an incidence rate of 116 per 100,000 population, Brazil 44 per 100,000, and Mexico 22 per 100,000 [1]. Costa Rica, Cuba, Jamaica, Puerto Rico, and Trinidad and Tobago have incidence rates between <10 and 20 cases per 100,000 people, values closer to the threshold for elimination (defined as <1 case per million) [8,9,10].

Throughout the Americas, drug resistance presents a particular crisis. In Peru, drug-resistant tuberculosis accounts for 9% of its cases, compared with 3% in each Brazil and Mexico [8]. In 2017, only 33% of patients received drug-susceptibility testing, resulting in an estimated 7000 patients with drug-resistant tuberculosis remaining undiagnosed or untreated [8]. Furthermore, the proportion of successful treatment outcomes for MDR-TB in the Americas in 2015 was 56%, with 26% lost to follow-up [8]. Comorbid infections are a challenge in Mexico where 30–40% of tuberculosis cases have diabetes [9,11,12], while large cities and incarcerated populations significantly contribute to new cases of disease in several Latin American countries, including Brazil and Peru [13].

### 2.2. The Genomic Sequencing of *Mycobacterium tuberculosis*

The application of genomic sequencing to the field of microbiology has improved our understanding of the evolution, diagnosis, and pathogen determinants of human disease. Since the MTB whole genome was sequenced in 1998 [14], new genomic techniques and technologies have led to remarkable improvements in accuracy, speed of processing, and analysis of samples alongside significant reductions in cost. Consequently, phylogenetic studies have revealed that the diversity of MTB, which forms part of the *M. tuberculosis* complex, is much greater than originally understood [15,16].

This genetic diversity has been exploited through a variety of methods to classify MTB into robust phylogenetic strains, leading to an emerging consensus which categorizes MTB into six main strain lineages [17]. Initially, researchers relied on standard epidemiological typing techniques such as insertion sequence 6110 restriction fragment length polymorphism (IS6110 RFLP), spoligotyping, and mycobacterial interspersed repetitive unit-variable number tandem repeats (MIRU-VNTR) for classifying MTB strains. IS6110 RFLP uses mobile DNA elements which are unique to the MTB complex to sub-speciate isolates [18,19]. Spoligotyping (or spacer oligonucleotide typing) is a rapid, polymerase chain reaction (PCR)-based method for genotyping members of the MTB complex. The technique employs primers to 43 spacer regions within the direct repeating (DR) region of the tuberculosis genome. The presence or absence of spacer hybridization provides a 43-digit binary output which can be converted into an octal code (or spoligotype) that is used to define the strain of MTB. MIRU-VNTR typing relies on counting the number of DR units by quantifying the size of PCR amplicons at 12, 15, or 24 loci in the MTB genome.

More recently, unique genetic markers have been identified and characterised in the form of large sequence polymorphisms (LSPs) and single nucleotide polymorphisms (SNPs) [17]. SNPs are phylogenetically useful mutations because the low genetic variability of the MTB complex makes independent recurrent mutation very unlikely, while the lack of horizontal gene transfer further reduces the likelihood of observing independent recurrent SNPs [17]. In their 2007 paper, Gagneux and Small reviewed the literature that had applied these phylogenetically informative mutations to a globally sampled collection of MTB isolates to define the phylogeny of *M. tuberculosis* and *M. africanum* (both species of the *M. tuberculosis* complex) into six main strain lineages, which are associated with particular geographical regions [17]. Within these lineages, several spoligotypes or strains exist.

### 2.3. The Arrival of Tuberculosis in Latin America

Tuberculosis has co-evolved with humans [20]. As MTB is an obligate human pathogen with long epidemic cycles over many years [21], much of the contemporary population diversity of MTB is defined by the origins of the first colonizing humans. It is hypothesized that the population density of indigenous pre-Columbian societies was sufficient to allow the spread of tuberculosis [22]. There is considerable evidence from pre-Columbian ceramic records that suggests mycobacterial disease was the cause of the characteristic thoracic gibbus deformity caused by Pott’s disease of the spine [23]. The ceramic record may however represent *Mycobacterium bovis*, *Mycobacterium pinipedii*, *Mycobacterium caprae*, or other members of the *M. tuberculosis* complex capable of causing Pott’s disease. The pre-Columbian indigenous communities of Latin America were ancestors of those that migrated across the Bering Strait and southward through North America [24]. Despite this, the population diversity of MTB in Peru, which had one of the largest pre-Columbian populations, is almost exclusively comprised of Euro-American lineage 4 genotypes, not northern Siberian genotypes that one might expect if MTB arrived in the country via the Bering Strait [25]. This is consistent with MTB being brought to Latin America by the conquistadores and not beforehand. In 2014, Bos et al. sampled the diseased vertebral columns of pre-Columbian mummies from three distinct geographical sites in Peru. Using whole-genome sequencing, they demonstrated that *M. pinipedii* was the causative organism in these cases, further supporting the hypothesis that MTB arrived in the Americas after Christopher Columbus [26]. More recently, a phylogenetic study of a global collection of lineage 4 Euro-American strains dated the arrival of the Euro-American lineage to Latin America after the year 1500 [27]. Until evidence is found of MTB *sensu strictu* in Pre-Columbian mummified remains, the evidence currently points towards the arrival of MTB with the conquistadores.

### 2.4. The Phylogenetics of Latin American Tuberculosis

Unlike many bacterial pathogens, horizontal gene transfer does not occur in MTB. Most mutation events involve deletion, duplication, insertion, and SNPs. MTB therefore has a clonal pattern of evolution, such that strains and lineages emerge from a common ancestor. Despite this, whole-genome sequencing has revealed substantial genetic diversity between strains, leading to an improved phylogenetic SNP-based strain classification [28]. Studies in Peru [25,29,30], Argentina [31], Mexico [32,33], and Brazil [34,35,36,37] have all examined the genetic diversity of MTB in Latin America and have established that the predominant strains have evolved from the Euro-American lineage 4. This domination of lineage 4 in South America, alongside its worldwide distribution, supports the hypothesis that it was introduced and dispersed by European colonialists in the mid-sixteenth to nineteenth centuries, as discussed above [38,39,40,41].

There are differences in genotype distributions between and within countries in Latin America (Figure 1, Table 1 and Table 2) that likely reflect further patterns of human migration. A 2018 systematic review by Wiens et al. looked at the global variation in MTB strains and found that Central America and northern South America had different genotype distributions from central and southern South America [38]. Due to historical Chinese immigration for work on guano factories and railways, Peru has a relatively higher number of Beijing strains than neighbouring countries, comprising 9% of the MTB population [25]. Furthermore, a study in Acapulco, Mexico found that the largest cluster identified, comprising 33.6% of isolates, was the Manila family, a member of the East African Indian group (EAI2). This is likely to reflect Mexico’s historic ties with Asia, as the Manila genotype is found throughout South East Asia, particularly the Philippines, Myanmar, Malaysia, Vietnam, and Thailand [42].

The differences in MTB genotypes throughout Latin America are likely to be the result of different periods of human migration and colonisation over time—for example in Brazil, with initial Portuguese colonisers and the forced migration of large numbers of African slaves in the 16th–18th centuries, followed by significant European immigration throughout the 19th and 20th centuries [43]. This may be the reason behind the development of the RD-Rio strain found disproportionately in Rio de Janeiro [44]. This contrasts with patterns of migration in other countries such as Argentina which experienced proportionately larger numbers of migrants from Europe and the Middle East [45,46]. These differences are likely to have contributed to the diverse phylogeography of MTB in Latin America that we see today.

There has also been migratory flux between countries within Latin America; in Chile, cases of Beijing clade MTB have been linked to recent immigration from countries such as Peru [47]. Ongoing migration, including the exodus of large numbers of people from Venezuela over recent years, is likely to have implications for the pattern of tuberculosis transmission in the future [48].

### 2.5. Genetics of *Mycobacterium tuberculosis* Disease

The clinical spectrum of tuberculosis is a consequence of the interaction between host factors, such as the host’s ability to clear or control infection, environmental factors such as the bacillary load and duration of exposure to infectious droplets, and pathogen factors, which are becoming increasingly understood. Identifying pathogen genetic biomarkers of virulence has potential implications for worldwide tuberculosis control, as they may provide targets for new treatments, vaccines, and diagnostic tools [64].

The complex interaction between host, environment, and bacterial factors results in difficulty independently attributing genotypic differences between strains to their phenotypic manifestations, thus, understanding the virulence of MTB has proved challenging [15]. Although there are multiple definitions used to define a virulence gene, a necessary characteristic is that inactivation of a candidate gene in the mycobacterial genome leads to a measurable loss of virulence in an in vivo model. From a clinical and epidemiological perspective, frequently used measures of virulence are morbidity and mortality, that is, the proportion of infected patients (or animals) that die, and the time taken to do so following infection.

In 2001, Manca et al. compared the virulence of two MTB clinical isolates following aerosol-mediated infection of immunocompetent mice. The increased virulence of the HN878 strain compared to the NHN5 strain was demonstrated through an accelerated time to death of mice infected with the HN878 strain [65]. Similar methods have been used to demonstrate the increased virulence of two “modern” Beijing strains isolated in Mozambique and Brazil compared with “ancient” strains [66]. This has been correlated clinically with findings that most modern strains come from environments where they have caused large numbers of clustered secondary cases in immunocompetent individuals, suggesting high transmissibility [66]. Additionally, in Japan, where ancient Beijing strains are prevalent, recently transmitted cases are more frequently caused by modern Beijing strains, whereas ancient strains are associated with reactivation in older patients [66].

The function of virulence genes can be classified broadly by molecular and biological roles, such as cell secretion and envelope functions, enzymes involved in cellular metabolism, and transcription regulators. Clinically and epidemiologically, these molecular and biological processes each contribute to the following aspects of mycobacterial virulence, as outlined by Coscolla and Gagneux: “(i) the ability of the bacteria to survive in the face of host immune responses, (ii) their capacity to cause lung damage, (iii) to survive the aerosolisation process outside of the host, and (iv) successfully transmit to and infect a new host” [67]. MTB has no other animal or environmental reservoir, and as such must cause pulmonary disease in order to be transmitted. In this way, tuberculosis virulence is directly linked to transmission [67,68,69].

It is currently estimated that over one hundred genes are associated with virulence in MTB. An exhaustive summary of these genes is outside the scope of this review; however, there are comprehensive reviews published on this topic [70,71].

### 2.6. Genetic Determinants of Disease Phenotype

Evidence increasingly suggests that MTB pathogen genotype is associated with distinct phenotypic features that may account for disease presentation and outcome. This idea emerged in the 1960s after studies in guinea pigs showed that MTB strains isolated in India and Thailand were less virulent than those isolated in the UK [16,72,73]. Epidemiological observations have demonstrated that there are a small proportion of strain genotypes which give rise to a disproportionately large number of cases, particularly in the case of a number of American outbreaks [74,75], and that some strains are associated with prolonged outbreaks [74,76,77]. More recently, in vitro immunogenicity and animal studies have shown that there are strain-specific differences in gene expression which are likely to result in phenotypic variation [16,78,79], and a few studies have revealed strain-specific differences in virulence and immunogenicity [16,80,81,82].

Until genomic techniques became more widely available in the 1990s, the dogma was that the genetic diversity of MTB was too narrow to account for differences in virulence, and could be accounted for by differences in host and environmental factors [16]. However, approximately two-thirds of coding SNPs in MTB are non-synonymous, meaning they result in a change in the encoded amino acid [16,83,84]. A large proportion of these (58%) are predicted to affect gene function, which suggests that the genetic differences observed among MTB strains have phenotypic consequences [16,40].

### 2.7. Ability to Cause Active and Cavitating Disease

One of the key mechanisms of MTB virulence is its ability to subvert the host immune response. This has been investigated using whole-genome sequencing of clinical MTB strains to understand the genetic diversity of human T cell epitopes [16,84]. While in most pathogens immune pressure drives evolution and antigenic diversity [85], it has been shown that human T cell epitopes among MTB strains are hyper-conserved. This suggests that the bacteria may benefit from recognition by the immune system and the immune responses that follow, possibly through the promotion of tissue damage and therefore enhanced transmission from patient to patient [16,84]. This is supported by the finding that cavitating disease is more likely to give rise to secondary cases [67,69].

Several studies have found that increased MTB virulence is associated with reduced or delayed inflammatory responses. The underlying mechanism could be that the initial reduced immune response allows bacterial proliferation earlier in the infective process, thereby leading to increased virulence at a later stage [67]. The NH878 strain, a member of lineage 2/Beijing, is consistently associated with a reduced inflammatory response and increased virulence [65,67,86,87,88,89,90,91,92]. Similarly, modern lineages 2 and 4 have been found to promote a reduced early inflammatory response compared to ancient lineages 1 and 6 [91]. These modern strains are far more widely distributed, and it has been postulated that their particular interaction with the human immune system may have resulted in their successful spread [93]. In fact, modern strains have been shown to replicate faster in vitro in both human monocyte-derived macrophages, and in aerosol-infected mice, perhaps indicating the survival advantage of a reduced early immune response [94]. However, studies comparing the anti-inflammatory phenotypes of different strains are conflicting. This may be due to differences in experimental conditions, or possibly because sub-lineages within the main lineages may vary significantly in this regard [67,95].

Another important virulence mechanism is the ability to cause active disease as opposed to latent infection or being cleared entirely by the immune system. A study in The Gambia has demonstrated that individuals infected with modern lineages 2 and 4 are more likely to progress to active disease compared to lineage 6 (though not at transmission level) [16,96]. However, a study in Ghana did not find the same association [97]. Additionally, various studies have shown that strains from lineages 2 and 4 cause more severe disease [16,98]. There may be variation within MTB lineages as well as between them.

Mechanisms other than subversion of the host immune response may account for lineage-specific differences in virulence. Variable virulence may arise by altering bacterial uptake by host cells, differences in cytokine induction, and intracellular growth [67,94].

In 2007, the RD-Rio clade was first described as being the predominant MTB clade in Rio de Janeiro, Brazil and is the clade most frequently associated with recent transmission [44]. The presence of these strains in a high incidence area could indicate a selective advantage in terms of virulence and transmission, an idea supported by evidence of a disproportionate number of clustered cases, a higher secondary case rate, and more cases in children, suggesting the effective spread of disease [44,55,99,100]. Furthermore, RD-Rio may cause a more severe form of illness, as evidenced by its tendency to cause increased weight loss, haemoptysis, and a higher bacterial load in sputum samples [44]. It has also been suggested that it could be associated with a cavitary disease phenotype and with drug resistance [36,101]. However, other studies have not supported this, suggesting that RD-Rio strains do not cause distinctive or more severe disease than non-RD-Rio strains [102].

In Argentina, a four-decade long outbreak of MDR-TB has been reported, caused by the M-strain. It has been estimated that it acquired resistance to isoniazid, rifampicin, and streptomycin by 1973, and additional resistance to six drugs in total by 1979. This strain demonstrated a remarkable ability to acquire drug-resistant mutations, despite a relatively slow mutation rate of 0.29 mutations per genome per year over the entire period of the outbreak. It was suggested that the ability of MTB to respond to the selective pressure of antibiotic therapy is the result of large within-host populations of bacteria. This allows the selection, and thus transmission, of resistant variants [31].

### 2.8. Ability to Transmit

Transmission of MTB largely depends on the ability of the pathogen to cause lung damage, and it therefore follows that genotypes predisposed to causing pulmonary or cavitary disease, as opposed to extra-pulmonary disease, will more efficiently transmit. Animal experiments in the marmoset model have demonstrated that lineage 6 more frequently spreads to lymph nodes, liver, and spleen compared to lineage 4 [98]. However, in humans, findings have been inconsistent, variably finding that lineage 2 is associated with extra-pulmonary TB [103,104,105], or that lineages 1 and 3 have this association when compared to lineage 2 [106], or that lineage 4 is associated with pulmonary TB as opposed to meningeal TB [107], or that there is no association at all between lineage and presentation [108,109,110,111,112].

There have been efforts to compare the transmissibility of MTB strains, through methods such as measuring changes in the proportion of certain genotypes in patient populations over time, comparing genotypic clustering in a particular setting, or studying genotypes associated with younger patient age (as TB in young people is more likely to represent ongoing transmission instead of reactivation) [113,114]. Through these methods, a view has developed that modern MTB lineages are more transmissible than others [67]. Lineage 2 has been reported to show higher genotypic clustering in Vietnam and Shanghai, and lineage 2/Beijing has been reported to be increasing in frequency over time in several different settings, although there have also been contradictory findings [115,116,117,118,119,120,121,122,123,124,125,126,127,128]. Furthermore, Wiens et al. found that strains thought to be more recently evolved (lineages 2, 3, and 4) were more widely distributed around the world [38]. This could be because older strains such as lineages 5 and 6, which are largely concentrated in West Africa, are better adapted to their local populations and do not spread as effectively beyond them [38]. In their 2014 review on the subject, Coscolla and Gagneux argued that there is evidence to support the view that on average, modern MTB lineages are more transmissible than others, although general patterns may be obscured by differences between sub-lineages [67].

In Peru, Grandjean et al. [25] demonstrated that Haarlem strains were associated with the occurrence of secondary cases of disease in a household context. This contrasts with evidence from Asia which suggests that Beijing strains are associated with an increased tendency to give rise to secondary cases of disease [129]. This could be explained by the predisposition of certain MTB sub-lineages to growth in specific host populations.

### 2.9. Phylogenetic Associations with Acquired Drug Resistance

Following exposure of selected strains of lineages 2 and 4 to drug pressure and performing before and after whole-genome sequencing, Ford et al. measured lineage-specific mutation rates [130]. They found that the mutation rate of lineage 2 strains was greater than that of lineage 4. It was concluded that this may explain the high prevalence of drug resistance observed in Eastern Europe. However, the strains chosen for comparison were not necessarily representative of the true population diversity of these lineages. The prevalence of drug resistance in Eastern Europe is also dependent on the pre-existing population diversity in the region (the founder effect). Prior to the collapse of the TB programs across the region in the 1980s, the pre-existing strains were predominantly lineage 2. Therefore, the prevalence and likelihood of transmission of drug-resistant Beijing strains may well have been higher, confounding this hypothesis. In Peru and across Latin America, in contrast to Eastern Europe, the prevalence of drug resistance is disproportionately found among the Latin American Mediterranean sub-lineage of Euro-American lineage 4 [25]. This too may be the result of the founder effect, or due to within-lineage differences in drug resistance acquisition.

## 3. Conclusions

Latin America has a varied tuberculosis landscape with hotspots of TB incidence in large cities and prisons, and with half of all disease accounted for by Peru, Brazil, and Mexico. New strategies must be sought to combat the spread of the disease. Understanding how pathogen genetic factors influence the transmission and virulence of MTB may help limit disease spread by enabling treatments and transmission-blocking interventions to be tailored to the bacterial genome. Early identification of patients who harbour highly transmissible strains could improve isolation practices, and transmission-blocking interventions, including latent TB therapy, could be cost-effectively applied to appropriate sub-groups of patients. Identifying pathogens most likely to become drug-resistant (or vice versa) would help to inform individualised management regimens, in terms of the choice of treatment and its duration.

Many studies examining the lineage-specific effects on clinical phenotypes have been substantially underpowered, have suffered from the potential misclassification of low-definition genotyping techniques, or have not used strains that represent the true population diversity of MTB. This has led to wide-ranging and varying conclusions about genotype–phenotype associations. Whole-genome sequencing is now inexpensive and rapidly implementable, making it possible to study phenotype–genotype correlations with much greater power and reproducibility.

Combining population level epidemiology, clinical variables, host genetics, and bacterial genetics will ultimately be required to determine pathogen genetic determinants of transmission in the field. The advent of gene editing techniques such as CRISPR (clustered regularly interspaced short palindromic repeats) in bacteria will enable the rapid generation and comparison of isogenic (otherwise identical) strains of MTB. This could then be used to study the virulence of candidate mutations against isogenic strains in animal models. Consequently, these strains could then be studied in the field, for example, by observing whether they are transmitted more frequently in close contacts compared to other strains.

Many countries in Latin America have implemented whole-genome sequencing as a diagnostic tool for drug resistance. The use of whole-genome sequencing to diagnose first and second line drug resistance has the potential to completely replace drug resistance testing by the laborious and time-consuming culture-based proportions method. Sequencing pipelines are now entirely automated enabling scarce human resources to be employed elsewhere in TB control. If used in real time, whole-genome sequencing also has the potential to rapidly detect new outbreaks with greater sensitivity and, in so doing, identify new sites of transmission. An estimated 81% of tuberculosis transmission occurs outside the home [131]. Rapidly identifying and then intervening at newly identified sites of community tuberculosis transmission would be a valuable method of diminishing global incidence in line with targets set by the END TB Strategy. Even greater impact on TB control will be achieved if the pathogen genomic determinants of tuberculosis transmission and disease can be elucidated. This will further individualise interventions and treatments to those most likely to transmit and cause widespread disease.

## Figures and Tables

**Figure 1 genes-10-00053-f001:**
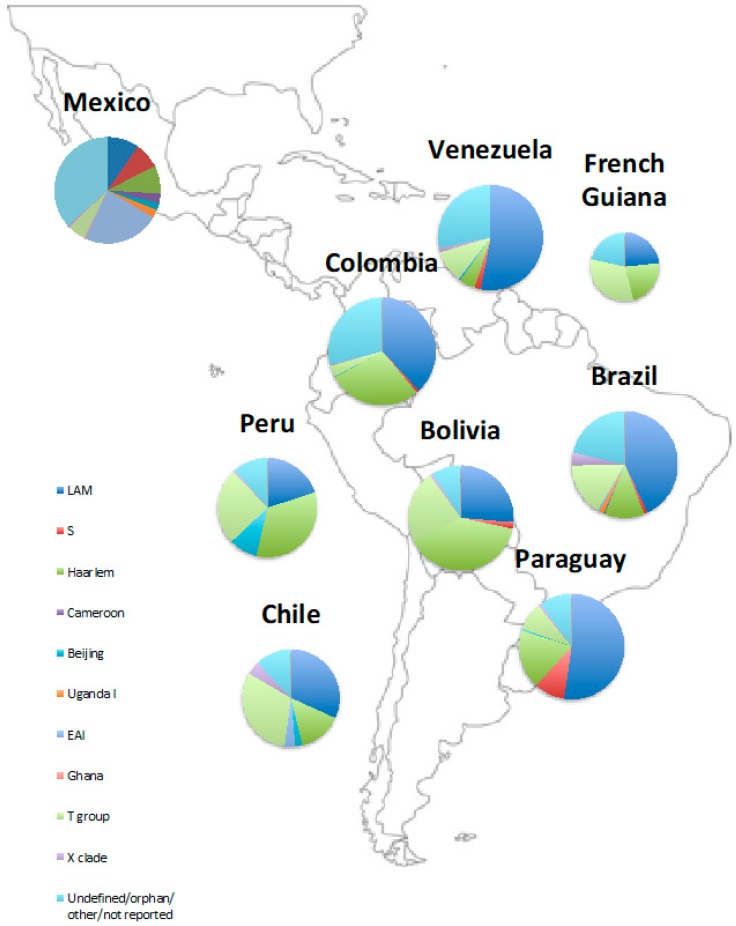
Distribution of phylogenetic groups in Latin America. References for data are included in Table 1 and Table 2.

**Table 1 genes-10-00053-t001:** The population diversity of Latin American MTB spoligotypes. The proportion of different spoligotypes in each country/region is expressed in percentages. Most studies did not report all spoligotypes found; their percentages therefore do not add up to 100%.

Country/Region	LAM (%)	S (%)	Haarlem (%)	Cameroon (%)	Beijing (%)	Uganda (%)	EAI (%)	Ghana (%)	T Group (%)	X Clade (%)	Undefined/Orphan/Other (%)
Acapulco, Mexico [42]	-	1.9	3	-	-	-	44.6	-	11.2	1.1	4.9
Baja California, Mexico [49]	19.3	13.6	14.3	5.7	4.3	5.0	2.1	0.7	-	-	35.0
Bogota, Colombia [50]	49.3	3.3	25.0	-	0.7	-	-	-	13.8	1.3	6.6
Rio Grande, Brazil [35]	54.0		16.0	-	-	-	-	-	22.0	-	-
Rio Grande, Brazil [34]	34.0	4.2	12.8	2.1	-	10.6	-	-	-	14.9	2.1
Rio de Janeiro, Brazil [37]	43.6		18.3	-	0.5	-	2.3	-	34.9	0.5	-
Rio de Janeiro, Porte Alegre, and Belem, Brazil [51]	66.2	2.5	9.7	-	0.5	-	3.0	-	2.0	5.2	10.1
Parana, Brazil [52]	26.9	-	17.2	-	-	-	-	-	11.8	-	-
Minas Gerais state, Brazil [53]	66.3	1.9	5.8	-	-	-	-	-	14.4	1.9	8.7
Sao Paulo, Brazil [54]	2.8	-	5.7	-	-	-	-	-	35.4	2.8	7.1
Southeast Brazil (prison population) [55]	50.0	-	11.5	-	-	-	-	-	8.7	5.7	9.2
Metropolitan region, Chile [56]	39.5	-	7.0	-	4.7	-	-	-	32.5	2.3	-
Peru [57]	23.8	-	23.8	-	9.3	-	-	-	22.3		-
Peru [58]	16.2	-	43.7	-	9.1	-	-	-	27.5	1.4	-
Santiago, Chile [47]	23.9	-	22.1	-	-	-	7.1	-	29.9	6.7	-
Medellin, Colombia [59]	39.6	-	48.7	-	-	-	-	-	-	-	6.4
Cali, Colombia [59]	39.1	-	39.0	-	-	-	-	-	-	-	11
Cauca State, Colombia [59]	24.0	-	-	-	-	-	-	-	-	-	40
French Guiana [60]	23.3	-	22.6	-	-	-	-	-	32.6	-	-
Bolivia [61]	26.3	2.0	39.4	-	-	-	-	-	22.2	1.0	9.1
Paraguay [62]	52.3	9.5	18.2	-	0.5	-	-	-	8.6	0.9	-
Venezuela [63]	53.0	1.9	5.0	-	0.4	-	0.2	-	10.0	1.2	-

**Table 2 genes-10-00053-t002:** The population density of South American MTB spoligotypes by country.

Country	LAM (%)	S (%)	Haarlem (%)	Cameroon (%)	Beijing (%)	Uganda I (%)	EAI (%)	Ghana (%)	T Group (%)	X Clade (%)	Undefined/Orphan/Other (%)
Mexico [42,49]	9.65	7.75	8.65	2.85	2.15	2.5	23.35	0.35	5.6	0.55	19.95
Brazil [34,35,37,51,52,53,54,55]	42.99	1.09	12.12	0.27	0.12	1.33	0.66	-	16.15	3.87	4.64
Peru [57,58]	20	-	33.73	-	9.22	-	-	-	24.88	0.71	-
Chile [47,56]	31.7	-	14.55	-	2.35	-	3.55	-	31.2	4.5	-
Colombia [50,59]	38.01	0.82	28.18	-	0.16	-	-	-	3.46	0.33	1.64
French Guiana [60]	23.3	-	22.6	-	-	-	-	-	32.6	-	-
Bolivia [61]	26.3	2	39.4	-	-	-	-	-	22.2	1	9.1
Paraguay [62]	52.3	9.5	18.2	-	0.5	-	-	-	8.6	0.9	-
Venezuela [63]	53	1.9	5	-	0.4	-	0.2	-	10	1.2	-

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
