# Peer review of "Tuberculosis Genetic Epidemiology: A Latin American Perspective"

_genes, 2019, doi:10.3390/genes10010053_

Round 1
Reviewer 1 Report
This review brings together some basic epidemiological data, and information on particular genetic events underpinning virulence, transmissibility and resistance. At this time, it would benefit from some rewrite to have a more effective message.
1. It would be useful to have a figure that summarizes basic epidemiological metrics for South America (now in text), as well as some of the novel data from available molecular epidemiology studies (eg., presence of lineages in specific regions)
2. The manuscript needs to introduce the concepts and definitions of what makes a lineage, a spolygotype, a “modern” lineage, as sub-lineage, clades, M-strains, RD-Rio strains etc.
3. Please have a more complete describe on how virulence transmissibility is assessed. Also, how do you assess transmissibility vs epidemiology setting (eg. a strain that spreads in prison is particularly transmissible, or just the result of an epidemiological opportunity).
4. The text is not very informative about actual pathogen genetics. It may be useful to have a table of text that actually identifies genes that have been proposed/confirmed to have a role in virulence, drug resistance, and other critical biological parameters
Other comments:
Line 33. The statement that follows refers specifically to MDR-TB, but it is written as having a general validity: “New therapeutic strategies are now available in many settings with bedaquiline (2), delamanid (3) and pretomanid (4). Injectable free regimes will also serve to significantly improve compliance and hopefully cure rate.”
Line 37-38: The statement: “…have already elucidated…….(6) and tuberculosis transmission measures” needs to be rephrase to either “tuberculosis transmission epidemiology” or to “ have already guided tuberculosis transmission measures “
Line 52: Typo: Puerto Rica
Line 77: Technically, you are referring to Mycobacterium tuberculosis complex (M. tuberculosis, M. bovis, M. bovis BCG, M. africanum, M. microti, M. canetti, М. pinipedii, M. caprae) – but the text is ambiguous by referring to Not-tuberculous mycobacteria.
Line 93. Needs to introduce the concept of Spoligotypes so the readers can understand the content of Table 1.
Line 253. The statement “The advent of CRISPR in bacteria will enable the study of candidate mutations against isogenic (otherwise identical strains) in animal models. This will then inform clinical trials in the field.” – Unclear ho CRISPR studies would influence a clinical trial in the field.
Reviewer 2 Report
After a revision of manuscript: “Tuberculosis Pathogen Genetics: A South American Perspective”, here are my comments. The goal of this document is to show the diversity of tuberculosis in South America, and how the variation observed could have implications in the clinical and epidemiological conditions of this disease in the region. The document is well written and shows the most important and actual information related with this disease in the region. However some aspect needs to be considered.
First at all, I really thank to will appreciate to the authors that clarify that Mexico is placed in North-America, not South America, or use the term Latin-America that include all the countries from north, center and south America, that share somehow similar language and culture.
One aspect, that is not mentioned in the document, is the importance of migratory flux between the countries of the region and how this could be an important aspect in the epidemiological development and future of TB. Could authors include a couple of lines about his important aspect?.
Results
Section 2.4 The phylogenetics of south America
It seems to me that this section requires a more in-depth analysis, although it has recently been established that the European conquerors were those who possibly introduced the various lineages that now predominate in the region, there are variations between countries that reflect their specific historical developments, for example the process of colonization in Brazil were completely different from that observed in Colombia, where you can see significant differences in the percentages of lineages (table 3). This is one of the substantial objectives of the manuscript but it is not sufficiently addressed.
In addition I believe that this article could be very usefully to reinforce the idea about the influence of the several cultures that drive the variation of lineages observed in specific regions from the Latin American countries. Clin Dev Immunol. 2011;2011:408375. doi: 10.1155/2011/408375.
This article, recently published, BMC Med. 2018 Oct 30;16(1):196. doi: 10.1186/s12916-018-1180-x. Show an excellent revision of the distribution of lineages and spoligotypes in different geographical regions, with emphasis in Latin America. Perhaps, could be useful to the authors, and helpful to confirm the distribution of the different lineages in Latin America. Here is the link.
Perhaps table 3 could be rearranged considering the country as principal element of aggrupation, perhaps this could evidence with more clarity, the differences between countries. Also a figure with the map of the Latin America region and the proportions of lineages found in each country could be very illustrative.
One major missing point in the final discussion, and that should be included, is the consideration of the impact that the WGS will have in the diagnostic, early identification of drug resistance profiles, the genotypic and phylogenetic characterization of TB, and how this could address the early diagnosis, individualized treatments and the potential impact the diminution of transmission of this disease, in a similar way to models described in England and European countries.
Round 2
Reviewer 1 Report
Thanks for the revisions that adequately answer to my queries